# NEURAL MALWARE CONTROL WITH DEEP REINFORCEMENT LEARNING

## ABSTRACT

Antimalware products are a key component in detecting malware attacks, and their engines typically execute unknown programs in a sandbox prior to running them on the native operating system. Files cannot be scanned indefinitely so the engine employs heuristics to determine when to halt execution. Previous research has investigated analyzing the sequence of system calls generated during this emulation process to predict if an unknown file is malicious, but these models require the emulation to be stopped after executing a fixed number of events from the beginning of the file. Also, these classifiers are not accurate enough to halt emulation in the middle of the file on their own. In this paper, we propose a novel algorithm which overcomes this limitation and learns the best time to halt the file's execution based on deep reinforcement learning (DRL). Because the new DRL-based system continues to emulate the unknown file until it can make a confident decision to stop, it prevents attackers from avoiding detection by initiating malicious activity after a fixed number of system calls. Results show that the proposed malware execution control model automatically halts emulation for 91.3% of the files earlier than heuristics employed by the engine. Furthermore, classifying the files at that time improves the true positive rate by 61.5%, at a false positive rate of 1%, compared to a baseline classifier.

## 1 INTRODUCTION

Malicious software, or *malware*, is a serious threat to computer users. As a first line of defense, users and organizations rely on commercial antimalware (*i.e.*, antivirus) products to detect malware on their computers, and the antimalware *engine* is a key component of these malware detection products. Prior to allowing an unknown file to be executed on the native operating system, the antimalware engine often tries to detect malware using two main approaches. First, static analysis employs malware "signatures" (*i.e.*, rules) to scan the unknown file to search for malicious byte sequences in the file without execution. Next, the engine utilizes one form of dynamic analysis called emulation to execute the file in a lightweight sandbox. Lightweight emulation does not analyze the unknown file in a full virtual machine (VM). Instead, the emulator mimics the response of a typical operating system. If the engine can detect malicious behavior during emulation, the antimalware system blocks the file from being executed on the native operating system and alerts the user that the file they are trying to install is malicious. As a result, the user's computer is not infected.

Previous dynamic analysis research has focused on analyzing the *sequence* of system application programming interface (API) calls made by the unknown file during emulation. Typically, the authors propose a recurrent, deep learning model to discriminate between the behavior of malicious and benign files. Pascanu et al. (2015) used a recurrent neural network (RNN), or an echo state network (ESN), in combination with either a logistic regression classifier or a multi-layer perceptron (MLP) to detect malware. Athiwaratkun & Stokes (2017) replace the RNN with a long short-term memory (LSTM) recurrent network or a gated recurrent unit (GRU), and they also propose a character-level convolutional neural network (CNN) to predict if an unknown file is malicious. A CNN followed by an LSTM is proposed for this task by Kolosnjaji et al. (2016). In all these solutions, the authors consider a fixed-length input buffer containing the events executed from the beginning of the file. The length of this pre-defined window, $\nu$, varies depending on the study with $\nu \in \{50, 100, 200, 65000\}$.

In this paper, we propose a novel algorithm, based on deep reinforcement learning (DRL), to overcome this limitation and learn the best time to halt the engine's emulation to predict whether the unknown

file is malicious or benign. DRL has been used previously to create adversarial samples to attack a malware classifier (Anderson et al. (2018)). Initially, we tried to train a classifier to halt emulation based on the previous $t_f$ (*e.g.*, 200) behavioral events, but the accuracy of this classifier was not sufficient to effectively halt the emulation. To the best of our knowledge, this is the *first* paper to propose using deep reinforcement learning to *protect* users from malware. This DRL-based neural network, combined with an event classifier and a file classifier, learns whether to halt emulation after enough state information has been observed or to continue emulation if more events are needed to make a highly confident prediction. Unlike previously proposed solutions, the DRL algorithm allows the engine to decide when to stop on a *per file* basis.

Results from analyzing a collection of malware and benign files demonstrates a significant improvement in the early stopping of the execution of the file. The DRL-based system halts execution of 91.3% of the files earlier than heuristics used by the production antimalware engine. When the execution is stopped by the DRL model, the true positive detection rate exhibits a relative increase of 61.5% at a false positive rate of 1.0% compared to the best performing baseline model proposed in Athiwaratkun & Stokes (2017). Our contributions include the following: 1) we propose a deep reinforcement learning-based system which predicts when to stop emulating an unknown file, 2) we show that the proposed system significantly outperforms several recent neural malware classification systems and 3) we provide a theorem and prove that the proposed DRL-based model outperforms these earlier baseline models for malware classification.

## 2 BACKGROUND

**Engine System Call Events.** The original data for our research was collected by scanning a large collection of Windows Portable Executable (PE) files with the *anonymized for submission* production antimalware engine. The engine collects behavioral events, $e_t$, observed during emulation. Most of the events are associated with APIs invoked during execution, but other behavioral events, such as unexpected instructions or constructs, are also captured. In our data, the engine records 114 event types, $e_t \in \{0, \cdots, 113\}$, ranging from file IO, registry APIs, networking APIs, thread or process creation and control, inter-process communication, timing, and debugging APIs. The behavioral events are logged using a special version of the antimalware engine, and the files are labeled with $L \in \{0, 1\}$ where 1 corresponds to a malware file and 0 indicates that the file is benign. Additional details on emulation and scanning can be found in Appendix A, and the threat model is given in Appendix B.

**Conventional Reinforcement Learning.** Conventional Reinforcement learning is normally formulated as a stochastic Markov Decision Process (MDP). There are four main components in a standard reinforcement learning structure, including States, Actions, Rewards and Policy. Each of them play a different role in formulating the RL environment. A general interpretation is that reinforcement learning is a technique to help an agent learn what is the best action and policy to take such that its expected rewards/penalties can be maximized/minimized under a stochastic MDP environment (Sutton & Barto (1998)). Conventional reinforcement learning definitions are provided in Appendix C.

## 3 NEURAL MALWARE CONTROL AND IMPROVED FILE CLASSIFICATION

An overview of the proposed DRL-based system is shown in Figure 1, and it has two main components: the *execution control model* and the *improved inference model*. An unknown file is emulated by the antimalware engine, and this generates a sequence of behavioral file events, $E$.

The **execution control model** processes $E$ and is responsible for controlling the file's execution. If the execution control model can make a confident decision that the file is either malicious or benign, the execution is halted. As it is received, each individual event $e_t$ is first processed by an *event classifier* which makes a prediction, $y_{e,t}$, indicating whether or not the most recent event history includes malicious activity. Initially, we tried halting the emulation based solely on the *event classifier's* output, but the classifier's accuracy was not sufficient to accomplish this task and motivated the need for *deep reinforcement learning*. Even though it is a weak signal, $y_{e,t}$ is used to construct a reward signal for the DRL model, which then produces the execution control signal, $h_t$, indicating if the file execution should be halted or allowed to continue.

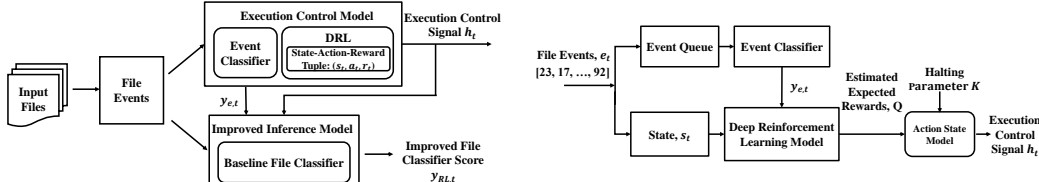

Figure 1: Deep reinforcement learning system for halting the execution of an unknown file and improved malware classification.

Figure 2: Details of the execution control model in Figure 1. In both figures, the input is $e_t$, and the outputs are $h_t$ and $y_{e,t}$.

The primary purpose of the DRL model is to better control the file's execution. However, we also found that it can be used to significantly improve the overall classification of an unknown file. This improved prediction, $y_{RL,t}$, which indicates whether the file is malicious or benign, is generated by the **improved inference model**. The improved inference model boosts the weak predictions from the *event classifier*, $y_{e,t}$, based on $h_t$ and the output of a *baseline file classifier* which offers an initial estimate of the probability, $y_f$, that the file is malicious based on the initial 200 events generated by the file.

## 4 DEEP EXECUTION CONTROL

The details of the execution control model in Figure 1 are depicted in Figure 2. The input is $e_t$, and the outputs are $h_t$ and $y_{e,t}$ in both figures. As each event is received, it is inserted into the *event queue*, a first in, first out (FIFO) queue. The event classifier then predicts $y_{e,t}$ for the most recent subsequence stored in the event queue. Since the output layer of the event classifier is a sigmoid function, $y_{e,t}$ is the probability that the most recent subsequence of behavioral events corresponds to malicious activity.

The DRL model depends upon its states, actions, and rewards. The state, $s_t$, includes information related to all the events received up to and including the most recent event. For each $e_t$, the event classifier's prediction $y_{e,t}$ is used as part of the DRL model's reward function, $r_t$. The actions for the DRL model include continuing and halting file execution. Based in part on $y_{e,t}$ and $s_t$, the DRL model generates separate $Q$ values which are the estimated expected discounted rewards associated with these actions. The $Q$ value signals are noisy and cannot be used directly. The *action state model* filters the $Q$ values to generate the halting signal $h_t$, which is then used by the antimalware engine to stop the file's execution.

**Event Classification**. The recurrent model structure which is used for the event classifier, and later for the baseline file classifier, is shown in Figure 3. For each new event, the event classifier makes a prediction, $y_{e,t}$, that the behavior associated with the most recent $t_f$ behavioral events is malicious. This event subsequence is stored in the event queue.

Each event in the event queue is input to an embedding layer, and the result is then input to a recurrent layer. We use a recurrent neural network (RNN) for the recurrent layer. As proposed in Pascanu et al. (2015), the recurrent layer's hidden state is input to a max-pool layer which is able to better detect malicious activity within the subsequence. We next construct a sparse binary, feature vector consisting of a bag of words (BOW) representation of the event subsequence (114), the final hidden state of the recurrent layer which is the recurrent layer's embedding (1500), and the output of the max-pool layer which is the max-pool embedding (1500). This feature vector is then input to a shallow neural network in the classifier layer. The output layer of the neural network is a sigmoid function. Thus, $y_{e,t}$ is the probability that this most recent event history contains malicious activity.

**Deep Reinforcement Learning.** We next present a deep reinforcement learning model to control the antimalware engine's execution of an unknown file. The first task is to choose the type of reinforcement learning model for our problem. The main feature in our problem is a very large state space together with a small action space consisting of two actions $A \in \{continue, halt\}$. Considering the small action space, we prefer a value-based reinforcement learning technique which

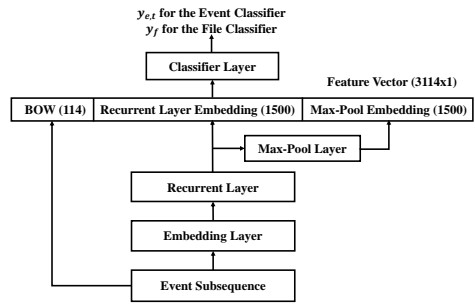

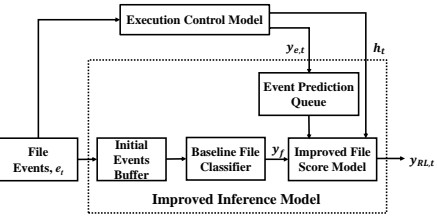

Figure 3: Structure of the *event classifier* and the *baseline file classifier*.

Figure 4: Details of the *improved file inference model* in Figure 1. The inputs are $e_t$, $y_{e,t}$, and $h_t$, and the output is $y_{RL,t}$ in both figures.

compares the value functions of the actions directly, instead of learning another policy estimator to find the best policy as in policy gradient (Silver et al. (2014); Levine et al. (2016)) or actor-critic-based approaches (Wawrzyński (2009); Wawrzyński & Tanwani (2013); Lillicrap et al. (2015)).

The action-value function, $Q^*(s_t, a_t) = \max_\pi \mathbb{E}[R_t|a_t, s_t, \pi]$, is the expected reward of taking action $a_t$ at state $s_t$ following the policy $\pi$. To calculate this value using a conventional value-based approach, it is necessary to store all the $Q$ values in a table for all the state-action pairs encountered during training, which is not feasible if the state space or action space is large.

One approach to overcome this difficulty is by training a nonlinear approximator $Q(s_t, a_t|\theta_t)$, such as a deep neural network, to estimate $Q^*(s_t, a_t)$ at time step $t$. However, these types of nonlinear estimators tend to be unstable in practical applications since convergence is not guaranteed. To address this issue, Mnih et al. (2013; 2015) recently proposed using a replay buffer in the deep $Q$ network (DQN) which demonstrates better convergence properties. Since our problem has a very large state space, we also use a DQN as our DRL model structure in this paper.

In a DQN-based DRL structure, the deep neural network action-value function estimator $Q(s_t, a_t|\theta_t)$ is normally defined at state $s_t$ as $Q(s_t, a_t|\theta_t) \sim Q^*(s_t, a_t)$ by taking the state $s_t$ as the input of the neural network, where $\theta_t$ represents the neural network parameters. We next describe the state design, action design, reward design, and training for the DRL model using experience replay.

**Design of States.** The DRL model's state, which is illustrated in Figure 5, contains three parts: the position (*i.e.*, index) of the current event in the file $\rho_t$, the current event ID, and the histogram of all the previous events. The current event position in the file will be used later to define the reward $r_t$ for the deep reinforcement learning model. We initially tried to use a one-hot encoding of the event ID, but found that using the event ID directly provides slightly better performance while reducing the size of the state. The event ID histogram captures the history of the events which have been observed so far.

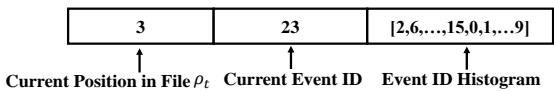

Figure 5: State representation, $s_t$, for the DRL model.

**Design of Actions.** The agent (*i.e.*, antimalware engine) can perform two types of actions $a_t$: **continue**, which is labeled as $C$, and **halt**, labeled as $H$. The action indicates whether the agent should continue or halt the execution of the file. In our deep reinforcement learning model, the selected action $a_t$ for state $s_t$ is inferred from the output of the neural network. As shown in the Figure 2, the outputs of the deep neural network are the estimated action value function $Q_C = Q^*(s_t, a_t = C) = \max_\pi \mathbb{E}[R_t|a_t = C, s_t, \pi]$ for action $C$ at state $s_t$ and $Q_H = Q^*(s_t, a_t = H) =$

$\max_\pi \mathbb{E}[R_t | a_t = H, s_t, \pi]$ for action $H$ at state $s_t$. By comparing the two $Q^*(s_t, a_t)$ values for actions $H$ and $C$, the action with the larger $Q$ value is selected and performed.

**Design of Rewards.** The reward $r_t$ at each state $s_t$ is designed based on two criteria:
1. We prefer for the DRL network to learn to halt emulation as quickly as possible. Therefore, shorter emulation sequence lengths are assigned a higher reward and longer sequence lengths are given a smaller reward.
2. The closer an event prediction is to the true label of the file, the larger the reward should be given at that state.

Based on the above two criteria, the reward is defined as

$$r_t = (0.5 - |y_{e,t} - L|)e^{-\beta \rho_t} \tag{1}$$

where $y_{e,t}$ is the event-based prediction generated by the most recent $t_f = 200$ events, and $L \in \{0, 1\}$ is defined as the true label of the training file. The decay factor $\beta$ is chosen experimentally, and $\rho_t$ is the position of current event in the file.

**DRL Training.** To train this neural network-based estimator, we use an $l_2$ loss function defined as $\mathcal{L}(\theta_t) = \mathbb{E}_{s_t}[(\hat{Q}(s_t, a_t | \theta_t) - Q(s_t, a_t | \theta_t))^2]$ where $\hat{Q}(s_t, a_t | \theta_t)$ is an estimate of $Q(s_t, a_t | \theta_t)$. $\hat{Q}(s_t, a_t | \theta_t)$ is computed using the current state reward $r_t$ together with its neighbors' estimations from the neural network in an iterative manner, *i.e.*, $\hat{Q}(s_t, a_t | \theta_t) = r_t + \gamma \max_{a_{t+1}} Q(s_{t+1}, a_{t+1} | \theta_t)$ where $s_{t+1}$ are the neighbors of $s_t$, and $a_{t+1}$ are the corresponding actions generated by the neural network.

In Mnih et al. (2013), experience replay is used to train the DRL model. Experience replay helps to alleviate the potential issues of non-stationary distributions and correlated data and is performed by randomly sampling the state pairs. Whenever one stochastic step is taken by the agent, the current state $s_t$, obtained reward $r_t$, action taken $a_t$ and next state $s_{t+1}$ are combined as one agent experience set $(s_t, r_t, a_t, s_{t+1})$, and pushed into the replay memory queue $M$. The reinforcement learning updates are performed in minibatches of size $B_{RL}$, drawing from the replay memory randomly. The algorithm for training the DRL model using experience replay is provided in Appendix D.

We tested several different stochastic gradient descent optimization methods for training the DRL model and found that adadelta (Zeiler (2012)) performed best. Furthermore, the convergence of DRL is not always guaranteed. To help with the convergence, the sum of $r_t + \gamma$ should be within the range of [0,1]. It is important to note that we first train the event classifier, as well as the *baseline file classifier* in the *improved inference model*, in isolation prior to training the DRL model – the system is not trained in an end-to-end fashion. Thus, $y_{e,t}$ in the reward $r_t$ is generated by the pre-trained event classifier, and the reward function has the same value for the same event sequence. Otherwise, the DRL's reward function can become non-stationary.

**Action State Model.** The final block in Figure 2 is the action state model which generates the halting signal $h_t$. The $Q$ value signal is noisy and cannot be used directly to compute $h_t$. The halting signal is a binary signal which is initialized to have a value of 0 and remains 0 for each $e_t$ until $Q_H > Q_C$ for $K$ consecutive events. At that point, the value of $h_t$ transitions to 1 and continues to maintain that value if any additional events are processed.

## 5 IMPROVED INFERENCE

The purpose of the *improved inference model* in Figure 1 is used to generate a better DRL-based prediction, $y_{RL,t}$, of whether or not an unknown file is malicious. The details of *improved inference model* are depicted in Figure 4. This model has three inputs, namely the events generated by the file ($e_t$), the most recent event predictions ($y_{e,t}$), and the execution control signal ($h_t$). The most recent $K$ values of $y_{e,t}$ are stored in the *event prediction queue*, which is another FIFO queue. After $h_t$ signals that the file execution has been halted, the *improved file score model* evaluates the event predictions in the queue to generate $y_{RL,t}$.

The individual $y_{e,t}$ values, stored in the *event prediction queue*, are noisy and can be difficult to analyze. In some cases, only setting $y_{RL,t}$ to be the most recent $y_{e,t}$ value can lead to an incorrect prediction that the file is malicious. To overcome this issue, the *improved inference model* also employs an additional *baseline file classifier* to improve the accuracy of $y_{RL,t}$. To accomplish this

task, the initial $t_f$ (*e.g.*, 200) events generated by the file are stored in the *initial events buffer*, and these are processed by the *baseline file classifier* to produce an initial prediction that the file is malicious, $y_f$.

**Baseline File Classifier.** The baseline file classifier also utilizes the structure shown in Figure 3 and follows Athiwaratkun & Stokes (2017). The input *event subsequence* in Figure 3 corresponds to the first $t_f$ events for each file stored in the *initial events buffer* (Figure 4). Here, $t_f$ is the same value which denotes the length of the event classifier's input *event queue*. An LSTM is used for the recurrent layer, and the classifier layer uses logistic regression for the file's prediction $y_f$. Similar to the event classifier, $y_f$ is the initial estimate of the probability that the file is malicious based on the initial behavior of the file.

**Improved File Score.** We can now combine $y_f$ with the $y_{e,t}$ history stored in the *event prediction queue* to compute the final improved file classifier score, $y_{RL,t}$. Since the initial estimates of the $y_{e,t}$ are noisy, we process the most recent $K$ event predictions from the event prediction queue. Formally if $h_t$ is equal to 1,

$$\begin{aligned} y_{RL,t} &= \max\{y_{e,t-K+1}, \cdots, y_{e,t}\} \quad \text{if } y_f > 0.5 \\ y_{RL,t} &= \min\{y_{e,t-K+1}, \cdots, y_{e,t}\} \quad \text{if } y_f \leq 0.5 \end{aligned} \quad (2)$$

where $y_{e,i}$ is the event classifier's prediction at step $i$, and $y_{RL,t}$ is the improved inference model's output, *i.e.*, the prediction probability that the unknown file is malicious.

It is important to provide mathematical insight why (2) provides better performance compared to the baseline model which only uses a deep learning-based classifier to process the first $t_f$ events. The intuition is that our new model provides a closer prediction to the true label (*e.g.*, malware or benign) compared to a single deep learning file classifier.

**Theorem 1.** *Let $\mathbb{E}_{s_{t_f}}[R_{t_f}]$ be the expected reward obtained at the state $s_{t_f}$ corresponding to event $e_{t_f}$, where $t_f$ is the fixed number of events used to train the baseline file classifier and is also the number of the most recent events used to train the event classifier. $\mathbb{E}_{s^*}[R^*]$ is the expected reward obtained at the halting state $s^*$ where $s^*$ is selected by the improved inference model when $h_t$ transitions from 0 to 1. If all models are fully trained and the decay factor $\beta$ defined in reward is small enough, $\mathbb{E}_{s_{t_f}}[R_{t_f}] \leq \mathbb{E}_{s^*}[R^*]$ and $|y^* - L| \leq \frac{1}{2}$ where $y^* \in [0,1]$ is the predicted probability of a file being malicious using the improved inference model, and $L$ is the true label of the given file. Then,*

$$\mid y_f - L \mid \geq \mid y^* - L \mid \quad (3)$$

*where $y_f$ is the predicted malware probability using a baseline file classifier.*

This theorem is proved in Appendix E.

Evasion of adversarial learning-based attacks is another important aspect to consider. Evasion of the model discussed in Appendix F.

## 6 EXPERIMENTAL RESULTS

We next present the results for the proposed neural malware control model. We first describe the datasets which were collected for our experiments. Next, we present the experimental setup. We then evaluate how quickly the DRL-based model halts the execution of a file. Finally, we compare the final prediction that the file is malicious or benign to the results from several baseline file classifiers.

**Datasets.** We collected a dataset of 75 thousand files which had been evaluated by a production antimalware engine. These files were equally split between the malware and benign classes. First, we discarded any files whose event sequences were shared between these two classes and which contained less than 50 events. Furthermore, we ensured that all the event sequences in the datasets were distinct. This requirement ensured that we did not overfit to one particular set of events. We then split the overall dataset into separate training, validation, and test sets with 50, 10, and 15 thousand files, respectively. Again, we maintained an equal split between the two classes for each of the individual datasets. The event and file classifiers were trained with the training and validation sets, while the DRL-model was trained with 2000 files from the training set. The results presented below are based on evaluating the model on the hold out test set.

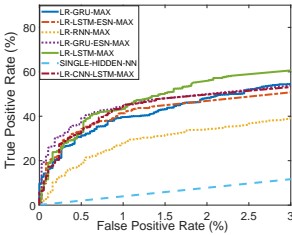
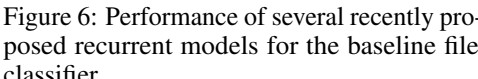
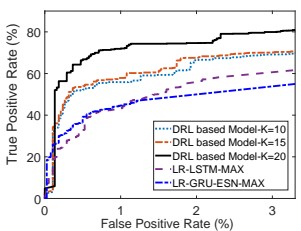

Figure 6: Performance of several recently proposed recurrent models for the baseline file classifier.

Figure 7: Comparison between the proposed DRL-based model for $K \in \{10, 15, 20\}$ and the best two baseline file classifiers.

**Setup.** We implemented the proposed neural malware control model using Keras (Chollet et al. (2015)) with Theano (Al-Rfou et al. (2016)) as the backend deep learning framework. Several hyperparameters were tuned on smaller datasets. The decay factor in the DRL model's reward function is $\gamma = 0.01$. The DRL model uses 3 hidden layers and is trained with a minibatch size $B_{RL} = 50$. The replay memory is initialized with a size $\mu = 50000$.

**How does the DRL-based model's stopping performance compare to the antimalware engine's heuristics?** The files in our dataset were collected by a production antimalware engine, and the number of events recorded for each file represents the performance of the heuristics employed by the engine to halt emulation. Thus, by measuring how often the DRL-based model halts execution prior to reaching the end of the file, we can compare the performance between our model and the engine's heuristics. In cases where the DRL model reaches the end of the file without halting execution, we can infer that the proposed model was not confident enough to make a decision, and the DRL based-model would have continued to execute the file.

The results of this evaluation are presented in Table 1 and depend on two values: the number of training files (*i.e.*, epochs) $N$ and the number of consecutive events where $Q_H > Q_C$ denoted by $K$. The fraction of files where the DRL-model halts execution before the end of the file, $\alpha$, is computed as: $\alpha$ = (Total number of early halted files)/(Total number of files).

We make two observations from the results presented in the table. First, the percentage of files whose execution is halted by the DRL model earlier than engine's heuristics continues to increase as the number of training file $N$ increases. Better training allows the engine to halt execution earlier. Second, the percentage of files which are halted early decreases with $K$. The value of $K$ is a proxy for the DRL model's confidence in the decision to halt the file's execution. It is not surprising that the execution of fewer files is halted early as we require more confidence (*i.e.*, higher value of $K$) in the decision. Even so, the results show that over 91% of the files in the test set are halted early compared to the engine's heuristics after training with only 2000 files. This indicates that the engine's heuristics may be overly cautious when emulating a file. In addition, requiring less time for scanning an individual file leads to better performance when scanning all the files on the hard drive. A histogram which indicates the percentage of events which were executed before the DRL model halted execution is provided in Appendix G.

|          | K=10  | K=15  | K=20  |
|----------|-------|-------|-------|
| $N$=30   | 71.5% | 64.1% | 58.3% |
| $N$=200  | 82.9% | 75.2% | 69.2% |
| $N$=2000 | 98.2% | 95.1% | 91.3% |

Table 1: The fraction of files, $\alpha$, in % where emulation is halted earlier by the proposed deep reinforcement learning model compared to heuristics used by the antimalware engine.

**Can DRL improve file classification?** While the results in Table 1 indicate that emulation of the majority of the files can be stopped earlier than the heuristics employed in the engine, it is important to understand how early halting affects the detection performance. To measure this, we first compute the receiver operating characteristic (ROC) curves for a range of models in Figure 6 for the baseline file classifier depicted in Figure 3. The models include LSTM Athiwaratkun & Stokes (2017), RNN Pascanu et al. (2015), gated recurrent unit (GRU) Athiwaratkun & Stokes (2017), convolutional

neural network (CNN) Kolosnjaji et al. (2016) and a simple, single hidden layer, feedforward neural network. We also include the echo state network (ESN) counterparts for the LSTM and GRU. All models use logistic regression (LR) for the classifier, because these slightly outperformed a shallow neural network for this dataset. All models use max-pooling as shown in Figure 3. These results indicate that the performance of none of the models we investigated dominated all of the other models. In particular, the ESN version of the GRU offers better performance and low false positive rates (FPRs) while the LSTM outperforms all other models above an FPR $\geq 1.2\%$.

We next compare the two best performing baseline file classifiers to the proposed DRL-based models for $K \in \{10, 15, 20\}$ in Figure 7. The figure clearly indicates that all the DRL-based models offer significantly better performance compared to the baseline file classifiers. In particular, the DRL-based model with $K = 20$ offers a relative improvement of $61.5\%$ for the true positive rate (TPR) at an FPR of $1\%$ compared to the GRU-ESN-based baseline file classifier. The relative improvement of the TPR is $65.7\%$ at an FPR of $1\%$ for the LSTM.

## 7 RELATED WORK

**Deep Reinforcement Learning.** Conventional reinforcement learning has been widely studied in the fields of machine learning and system control for over three decades (Sutton (1984); Williams (1992); Littman (1994); Kaelbling et al. (1996); Sutton & Barto (1998)). Recently, Mnih et al. (2013; 2015) successfully applied deep neural network-based Q-learning (DQN) to playing a series of Atari games, by using a replay buffer to improve the system's convergence. Also, Silver et al. (2016; 2017) developed novel algorithms, by applying reinforcement learning to Monte Carlo tree search, to play the Go game and beat human Go masters. Progress has also been made in improving value-based (Van Hasselt et al. (2016); Wang et al. (2016)), policy gradient (Schulman et al. (2015); Duan et al. (2016); Gu et al. (2017); O'Donoghue et al. (2017)), and actor-critic (Lillicrap et al. (2015); Mnih et al. (2016)) deep reinforcement learning algorithms, in order to find better policies more efficiently and to deal with a continuous action space.

Despite the success of applying DRL to different types of games, no existing research has studied malware control and behavioral classification using reinforcement learning. The proposed neural malware control model is the first to apply a deep reinforcement learning-based algorithm for these tasks.

**Deep Learning for Malware Classification.** A number of authors have proposed using DNNs for malware classification tasks. Kephart (1994) employed a shallow neural network in the first paper on malware classification. Dahl et al. (2013) first investigated the use of a DNN for malware classification for dynamic analysis. Huang & Stokes (2016) proposed a DNN with multitask learning for dynamic analysis where the first task was binary (*i.e.*, malware versus benign) and the second task was malware family classification. Saxe & Berlin (2015) proposed a DNN for the static analysis of malware. A separate line of research has investigated using recurrent models for malware classification. Pascanu et al. (2015) first proposed using recurrent neural networks and echo state networks for classifying malware sequences. Athiwaratkun & Stokes (2017) instead proposed an LSTM, a GRU, and a character-level CNN for the sequence classification. Kolosnjaji et al. (2016) used a CNN followed by an LSTM for this task.

## 8 CONCLUSION

We present a novel, neural malware control model which learns when to halt the execution of an unknown file based on deep reinforcement learning. This model is the first to use deep reinforcement learning to protect customers from malware. Fast scanning is an important feature for users when installing software, reading emails with attachments, and searching a hard drive for files which were maliciously dropped during a drive-by download. Our results indicate that the proposed model halts execution earlier than a production antimalware engine for more than $91\%$ of the files in the test set. More importantly, we show a relative improvement of over $61\%$ in the true positive detection rate of malware at a false positive rate of $1\%$ compared to a number of baseline malware classifiers reported in the literature. Thus, the proposed model offers significantly better protection with less delay.

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

APPENDIX A: EMULATION AND SCANNING DETAILS

In some cases, unknown files are evaluated in a protected sandbox environment to determine if they are malicious or benign prior to running them on the native operating system. This method of testing unknown files is called dynamic analysis, and there are two main modes of performing dynamic analysis: virtualization and emulation.

Virtualization involves running an unknown file in an instrumented version of a virtual machine (VM) which implements a full operating system. Performing analysis in a VM is the preferred method of analysis because it can fully implement all of the functionality required by the unknown file. Virtualization is often used in backend security services to protect customers. Advanced email services scan email attachments before allowing them to be displayed in the user's inbox. Antimalware companies also scan unknown files in full VMs in order to create new signatures and labeled malware files for training machine learning classifiers.

Emulation is often performed in antimalware engines running on the endpoint computer or cell phone. Emulation is similar to virtualization in that it tries to induce an unknown malware file into revealing its malicious behavior. However, emulation tends to be more restricted because it needs to run much faster and consume less resources compared to the execution of the file in a virtual machine. Thus, emulation can also be run in backend services to quickly analyze significantly larger numbers of unknown files compared to virtualization. If the antimalware engine does not detect that a file is malicious during emulation, it is then allowed to be installed and executed on the native operating system.

The data analyzed in this study was generated using emulation in a production antimalware engine. This data was generated in a production emulation scanning environment of a major antimalware computer, instead of by client computers located in the wild. The behavior revealed in our data is similar to that which would be encountered by the file running on the client computer's operating system except for one main difference. The antimalware engine does not allow the files to have internet access because this may lead to the infection of other computers in the network.

For emulation or virtualization, the behavioral event sequence corresponding to the system APIs called by the file are typically recorded. Attackers often use polymorphic tactics to avoid detection. In polymorphism, they rearrange or rewrite their code in different ways which appear to be different but accomplish the same task. To deal with polymorphism, our antimalware engine maps multiple low-level API calls into a single high-level event. For example, the attacker may use a user mode API (CreateFile), a kernel mode API (ZwCreateFile), or the C++ API (ofstream::open) to create a file. All of these events are mapped into the same high-level FileCreate event. As a result, the 114 behavior events represent many individual low-level API calls.

The labels utilized in this study correspond to production labels used by our antimalware product partners to train malware classifiers that identify malware targeting Windows computers. Benign files are identified as those which are known to be safe. For example, these might be files belonging to software products which are purchased by users (*e.g.* Microsoft Office) or downloaded from the internet from sites which are known to be legitimate (*e.g.* Adobe Acrobat Reader, Google Chrome). Other benign files are determined to be safe by professional analysts. Labels for malware files are also generated by manual inspection by professional analysts. In addition, all unknown files received by the company are scanned by over 20 additional production antimalware engines. If eight or more of these anti-malware engines detect that a file is malicious, these files will also be determined to be malicious.

APPENDIX B: THREAT MODEL

We next specify the threat model which defines the assumptions which are made about the proposed detection system and the attacker. For our system, we require that the computer has not been previously infected, and the trusted computing boundary includes the user account, operating system, and antimalware detection system. Otherwise, the attacker can alter the detection system's results and avoid detection. We also assume that the behavior observed in the antimalware engine's emulator without internet access is similar to that observed when the file is executed on the native operating system of the user's computer. Some malware tries to identify whether or not it is being emulated or virtualized. If the malware detects it is being emulated or virtualized, it may disable any further

malicious activity (*i.e.*, cloaking). Based on our current datasets, our system may fail to detect any malware which successfully employs these cloaking mechanisms.

## APPENDIX C: CONVENTIONAL REINFORCEMENT LEARNING DEFINITIONS

In this appendix, we provide definitions of the elements in conventional reinforcement learning.

1. **Agent and States ($s_t$).** An agent interacts with its environment by moving from the current state $s_t$ at time $t$ to another state $s_{t+1}$ at time $t + 1$. Each state is normally defined by the useful information from the interaction between an agent and its environment.

2. **Actions ($a_t$).** By taking an action $a_t$ at state $s_t$, an agent can transfer from its current state to any of its connected neighbors at its next state $s_{t+1}$ with different probabilities, since the agent can only arrive at one of its neighbors at $t + 1$.

3. **Rewards ($r_t$).** The agent receives reward $r_t$ at time $t$. The discounted reward $R_t$ is defined as $R_t = \sum_{t=t_0}^{\infty} \gamma^{t-t_0} r_t$ where $\gamma$ is the discount factor with $\{0 \leq \gamma \leq 1\}$, and $t_0$ indicates the starting time step. After reaching a state, an agent obtains the expected discounted reward ($\mathbb{E}[R_t|a_t, s_t]$) by considering the policies from the current state $s_t$ to its neighbors $s_{t+1}$ and so on. The expected discounted reward includes both the pre-defined reward at state $s_t$ and the accumulated discounted rewards to be obtained in the future by taking a specific action $a_t$.

4. **Policy ($\pi$).** A policy $\pi$ is a mapping from states to actions. There are three main types of reinforcement learning: value-based, policy gradient and actor-critic. In this paper, we focus on a value-based algorithm called Q-learning given the small action space in our problem. The optimal action-value function in conventional Q-learning is defined as $Q^*$, which is the maximum expected reward obtained by selecting the best policy $\pi$ at state $s_t$

$$Q^*(s_t, a_t) = \max_{\pi} \mathbb{E}[R_t|a_t, s_t, \pi]. \tag{4}$$

## APPENDIX D: A DRL ALGORITHM USING EXPERIENCE REPLAY

The detailed explanation of the DRL algorithm using experience replay in section 4.2 is provided in Algorithm 1.

## APPENDIX E: PROOF OF THEOREM 1

We present a detailed proof of Theorem 1 in this appendix.

*Proof.* We begin by assuming that both the baseline file classifier and event classifier are fully trained. By definition, the expected reward $\mathbb{E}_{s_{t_f}}[R_{t_f}]$ obtained at the state $s_{t_f}$ corresponding to event $e_{t_f}$ can be calculated as

$$\mathbb{E}_{s_{t_f}}[R_{t_f}] = p_{t_f} r_{t_f} + (1 - p_{t_f})\mathbb{E}_{s_{t_f+1}}[R_{t_f+1}] \tag{5}$$

where $t_f$ is the fixed number of events processed by the baseline file classifier, $p_{t_f}$ is the probability of remaining at state $s_{t_f}$, and $\mathbb{E}_{s_{t_f+1}}[R_{t_f+1}]$ represents the expected reward to be obtained by leaving $s_{t_f}$ for its neighboring state $s_{t_f+1}$ including the $t_f + 1$ event. After substituting the reward (1),

$$\mathbb{E}_{s_{t_f}}[R_{t_f}] = p_{t_f}(0.5 - |y_{e,t_f} - L|)e^{-\beta\rho_{t_f}} + (1 - p_{t_f})\mathbb{E}_{s_{t_f+1}}[R_{t_f+1}] \tag{6}$$

where $y_{e,t_f}$ is the predicted probability generated by the event classifier using events $[e_1, \cdots, e_{t_f}]$ (assuming the file has more than $t_f$ events). Similarly, $\mathbb{E}_{s_{t_f+1}}[R_{t_f+1}]$ can be written as

$$\mathbb{E}_{s_{t_f+1}}[R_{t_f+1}] = p_{t_f+1}(0.5 - |y_{e,t_f+1} - L|)e^{-\beta\rho_{t_f+1}} + C_{t_f+1} \tag{7}$$

where $C_{t_f+1} = (1 - p_{t_f+1})\mathbb{E}_{s_{t_f+2}}[R_{t_f+2}]$.

By choosing a small value of $\beta$, we may approximate that $e^{-\beta\rho_{t_f}} \approx e^{-\beta\rho_{t_f+1}}$ since $\rho_{t_f+1} = \rho_{t_f} + 1$. The event classifier should yield similar predicted probabilities for state $s_{t_f}$ and $s_{t_f+1}$ in a given file

---

**Algorithm 1** Deep Reinforcement Learning Training

---

1: Epochs: $N \leftarrow 2000$
2: Minibatch Size: $B_{RL} \leftarrow 50$
3: Decay Factor: $\beta \leftarrow 0.01$
4: Initialize a replay memory $M$ with size $\mu \leftarrow 50000$, DRL model with 3 layers
5: **for** n=1 $\rightarrow$ N **do**
6:     Time step in state space: $t \leftarrow 0$
7:     Randomly select an initial state $s_t$
8:     **while** !End of File **do**
9:         $Q(s_t, a_t|\theta_t) \leftarrow \text{DRL}(s_t)$
10:        $a_t^* = \arg\max_{a_t} Q(s_t, a_t|\theta_t)$
11:        Perform action $a_t^*$, generating next state $s_{t+1}$
12:        Push tuple $(s_t, r_t, a_t^*, s_{t+1})$ into replay memory $M$
13:        **for** b=1 $\rightarrow B_{RL}$ **do**
14:           Randomly select a tuple $m$ from $M$
15:           $s_t \leftarrow m(0), r_t \leftarrow m(1), s_{t+1} \leftarrow m(3)$
16:           $Q(s_t, a_t|\theta_t) \leftarrow \text{DRL}(s_t)$
17:           $Q(s_{t+1}, a_{t+1}|\theta_t) \leftarrow \text{DRL}(s_{t+1})$
18:           Input $y_{e,t}$ from Event Classifier
19:           $r_t \leftarrow (0.5 - |y_{e,t} - L|) \times e^{-\beta \rho_t}$
20:           Update $\hat{Q}(s_t, a_t|\theta_t)$
21:           Update the network by minimizing loss $\mathcal{L}(\theta_t)$
22:        **end for**
23:        $t \leftarrow t + 1$
24:     **end while**
25: **end for**

---

if the classifier is fully trained, *i.e.* $y_{e,t_f} \approx y_{e,t_f+1}$. Using these two approximations and substituting (7) into (6), (6) can be rewritten as

$$\mathbb{E}_{s_{t_f}}[R_{t_f}] = (p_{t_f} + (1 - p_{t_f})p_{t_f+1})((0.5 - |y_{e,t_f} - L|)e^{-\beta \rho_{t_f}}) + (1 - p_{t_f})C_{t_f+1}. \quad (8)$$

Since there are only two possible actions (halt, continue), the probability of remaining at a particular state $p_t$ at step $t$ is either 0 or 1. If the system halts before $t_f$ events, $p_{t_f} = p_{t_f+1} = 1$. Similarly, if the system continues until at least $t_f + 1$ events, $p_{t_f} = p_{t_f+1} = 0$. Assuming the system does not transition from continue to halt at event $t = t_f$, $p_{t_f} + (1 - p_{t_f})p_{t_f+1} = 2p_{t_f} - p_{t_f}^2 = p_{t_f}$. Under this weak assumption, (8) can be rewritten as

$$\mathbb{E}_{s_{t_f}}[R_{t_f}] = p_{t_f}(0.5 - |y_{e,t_f} - L|)e^{-\beta \rho_{t_f}} + (1 - p_{t_f})C_{t_f+1}. \quad (9)$$

Similarly, $\mathbb{E}_{s^*}[R^*]$, the expected rewards obtained at the halting state $s^*$ chosen by the DRL-based model, can be calculated as

$$\mathbb{E}_{s^*}[R^*] = p^*(0.5 - |y^* - L|)e^{-\beta \rho^*} + (1 - p^*)\mathbb{E}_{s^{*'}}[R^{*'}] \quad (10)$$

where $p^*$ is the probability of remaining at state $s^*$, $y^*$ is the predicted probability generated by the event classifier using the most recent $t_f$ events before the halting state $s^*$, $\rho^*$ is the position of the event contained in state $s^*$, and $\mathbb{E}_{s^{*'}}[R^{*'}]$ is the expected reward at state $s^{*'}$ which is the next state after $s^*$. Since $s^*$ is an absorbing (*i.e.*, terminal) state, $p^* = 1$. Thus,

$$\mathbb{E}_{s^*}[R^*] = (0.5 - |y^* - L|)e^{-\beta \rho^*}, \quad (11)$$

and the expected reward at $s^*$ is larger or equal than those at any other states

$$\mathbb{E}_{s_{t_f}}[R_{t_f}] \leq \mathbb{E}_{s^*}[R^*]. \quad (12)$$

Substituting (9) and (11) into (12),

$$p_{t_f}((0.5 - |y_{e,t_f} - L|)e^{-\beta \rho_{t_f}}) + (1 - p_{t_f})C_{t_f+1} \leq (0.5 - |y^* - L|)e^{-\beta \rho^*}. \quad (13)$$

Recalling $p_{t_f}$ is equal to either 0 or 1 and first considering the case when $p_{t_f}$ is equal to 1, (13) can be further rewritten as

$$(0.5 - |y_{e,t_f} - L|)e^{-\beta \rho_{t_f}} \leq (0.5 - |y^* - L|)e^{-\beta \rho^*}$$
$$0.5 - |y_{e,t_f} - L| \leq (0.5 - |y^* - L|)e^{\beta(\rho_{t_f} - \rho^*)} \tag{14}$$

Recall that the decay factor $\beta$ is defined to be positive. If $\rho_{t_f} < \rho^*$, then $0 < e^{\beta(\rho_{t_f} - \rho^*)} < 1$, and (14) can be rewritten as

$$0.5 - |y_{e,t_f} - L| \leq 0.5 - |y^* - L|$$
$$|y_{e,t_f} - L| \geq |y^* - L|. \tag{15}$$

If $\rho_{t_f} > \rho^*$, then $e^{\beta(\rho_{t_f} - \rho^*)} > 1$, and (14) can be rewritten as

$$0.5 - |y_{e,t_f} - L| \leq (0.5 - |y^* - L|)e^{\beta(\rho_{t_f} - \rho^*)}$$
$$\leq 0.5 - |y^* - L| + (e^{\beta(\rho_{t_f} - \rho^*)} - 1)(0.5 - |y^* - L|). \tag{16}$$

By choosing small enough $\beta$ such that $e^{\beta(\rho_{t_f} - \rho^*)} \to 1$, the second term in (16) is approaching zero, then $|y_{e,t_f} - L| \geq |y^* - L|$. To prove it strictly, choose a $\beta$ satisfying

$$(e^{\beta(\rho_{t_f} - \rho^*)} - 1)(0.5 - |y^* - L|) \leq \frac{1}{2}((0.5 - |y_{e,t_f} - L|) - (0.5 - |y^* - L|)). \tag{17}$$

By substituting (17) into (16), we can derive that $|y_{e,t_f} - L| \geq |y^* - L|$.

Given that $|y^* - L| \leq \frac{1}{2}$, (17) can provide an upper bound of $\beta$ as

$$(e^{\beta(\rho_{t_f} - \rho^*)} - 1)(0.5 - |y^* - L|) \leq \frac{1}{2}((0.5 - |y_{e,t_f} - L|) - (0.5 - |y^* - L|))$$
$$(e^{\beta(\rho_{t_f} - \rho^*)} - 1) \leq \frac{(|y^* - L| - |y_{e,t_f} - L|)}{2(0.5 - |y^* - L|)}$$
$$e^{\beta(\rho_{t_f} - \rho^*)} \leq 1 + \frac{(|y^* - L| - |y_{e,t_f} - L|)}{1 - 2|y^* - L|} \tag{18}$$
$$\beta \leq \frac{1}{\rho_{t_f} - \rho^*} \ln(1 + \frac{(|y^* - L| - |y_{e,t_f} - L|)}{1 - 2|y^* - L|}).$$

The proceeding derivation was for the case when $p_{t_f} = 1$. Following (5) for the case when $p_{t_f} = 0$, we have $\mathbb{E}_{s_{t_f}}[R_{t_f}] = \mathbb{E}_{s_{t_f+1}}[R_{t_f+1}]$. Similarly we have $\mathbb{E}_{s_{t_f+1}}[R_{t_f+1}] = \mathbb{E}_{s_{t_f+2}}[R_{t_f+2}]$ since $p_{t_f+1} = 0$, and so on. Thus, $\mathbb{E}_{s_{t_f}}[R_{t_f}] = \mathbb{E}_{s_{t_f+n}}[R_{t_f+n}]$ where $s_{t_f+n}$ is the first halting state after $s_{t_f}$, such that $p_{t_f+n}$ is equal to 1. Then we can substitute $t_f$ by $t_f + n$ in our proof in order to show the same result that $|y_{e,t_f} - L| \geq |y^* - L|$ for the case when $p_{t_f} = 0$.

While we have now shown that $|y_{e,t_f} - L| \geq |y^* - L|$ for the event classifier, we need to consider the prediction probability of the baseline file classifier. The predicted probability $y_{e,t_f}$ generated by the event classifier using the first $t_f$ events is an approximation of $y_f$, which is the predicted probability generated by the baseline file classifier also using the first $t_f$ events, i.e. $y_{e,t_f} \approx y_f$, hence we have

$$| y_f - L | \approx |y_{e,t_f} - L| \geq | y^* - L |. \tag{19}$$

It can be concluded that the DRL predicted probability $y^*$ is closer to the true label than the baseline file classifier's predicted probability $y_f$. $\qquad \square$

## APPENDIX F: EVASION

Recently, researchers have begun to investigate adversarial attacks on deep reinforcement learning models. Since the proposed DRL-based system would be run on a user's computer or cell phone, it is particularly vulnerable to adversarial attacks. The basic strategy is to destabilize the DRL network by

adding a perturbation to a state using a second adversarial agent (Lin et al. (2017); Pinto et al. (2017); Huang et al. (2017)). This research considers two main scenarios:

1. The attacker knows both the architecture and parameters of a trained neural network policy $\pi$. Thus, they can compute a perturbation directly based on the policy's parameters.

2. The attack is a black-box problem assuming the adversary has no access to the target policy, the training algorithm or the model's parameters.

One possible defense against adversarial attacks on the client is to either run the DRL model, or the entire antimalware engine, in a secure enclave, such as SGX (Ohrimenko et al. (2016)). Doing so prevents the attacker from directly observing the model structure and parameters, and the attacker must resort to a black-box type of attack (Huang et al. (2017)). Thus, the only way to build an adversarial agent is through the transferability across training algorithms, by guessing the target policy $\pi$ from another policy trained using a different dataset.

Protecting a DRL-model running outside of a secure enclave from adversarial attacks is an open research topic. Some promising approaches include the ensemble defense, feature squeezing or building a detection model to discover attacks (Tramer et al. (2018); Zantedeschi et al. (2017); Ilyas et al. (2017)).

## APPENDIX G: EARLY STOPPING DISTRIBUTION

In Table 1 in Section 6, we report that with $K = 20$ and $N = 2000$, the execution control model halts the execution of 91.3% of the files earlier than the heuristics used by the antimalware engine. In Figure 8, we provide the histogram indicating the distribution of the percentage of events which are executed by the engine prior to the execution control model halting its execution. This figure indicates that execution is halted much earlier than the heuristics for the majority of the files.

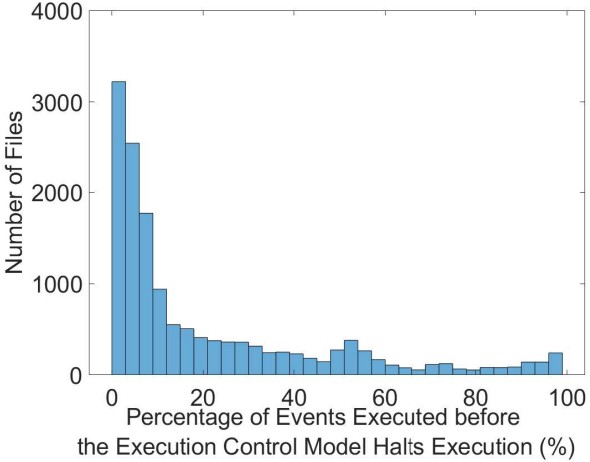

Figure 8: Histogram of the percentage of behavioral events which are executed before the execution control model halts emulation.

