# OpenReview forum: "NEURAL MALWARE CONTROL WITH DEEP REINFORCEMENT LEARNING"
_ICLR.cc/2019/Conference_

### Official Review · AnonReviewer2 · 2018-11-03
**using REINFORCEMENT LEARNING for NEURAL MALWARE CONTROL**

**Rating:** 5
**Confidence:** 2

**Review:**

This paper uses deep reinforcement learning (DRL) for malware detection. It can get better performance than LSTM or GRU based models.

Deep reinforcement learning (DRL) has already used for classification or detection. I am not sure about the main contribution of this work. The new application of DRL can not convince me.

As the dataset is not a public dataset, it is difficult to evaluate the performance. As for the comparing models, i think some CNN based methods should be included. If the task is a detection, i think some attention methods should also be investigated and compared. LSTM combined with attention should already be well investigated in other classification/detection tasks.

---

> ### Author Response · Authors · 2018-11-26
> **Response to AnonReviewer2**
>
> Thank you very much for reviewing the paper and your helpful feedback.
> 1. We believe this paper is important because it shows another “real-world” application of DRL on a very important problem in the security field. To the best of our knowledge, it is also the first study in security field to detect the malware using a DRL-based system. The originality of this paper is from a novel neural malware control model which learns the best time to halt the execution of an unknown file based on deep reinforcement learning. Also, our model contains several inter-connected submodules, and the DRL model is one of the important modules in the system. The focus of this paper is not an improved DRL algorithm, but to propose a novel system-level design using both DRL and DNN models to protect users from malware. Using DRL offers significant improvement (improves the true positive rate by 61.5%, at a false positive rate of 1%) compared to the best previously proposed solution.
> Another contribution of this paper is the unique reward design for this specific problem, the choice of rewards is not arbitrary but follows the two rules as described in paper: a) the DRL network can learn to halt emulation as quickly as possible.  b) The closer an event prediction is to the true label of the file, the larger the reward should be given at that state. Only because the reward is designed following these two rules, better results for malware detection are obtained. It is also the use of deep reinforcement learning that gives us the flexibility to design the rewards satisfying our requirements.
>
> 2. As suggested by the reviewer, we added one more experiment using a CNN-based classifier in Figure 6. Also, in (Athiwaratkun & Stokes (2017)), it was shown that the model with an attention layer performs worse than our baseline model with an LSTM and a max pooling layer on this malware classification task. For this reason, we did not compare the models with the one with an attention layer in our experiments. The max pooling layer seems to work much better due to sporadic nature of the malicious activity in malware.
> There are indeed many other models which provide good performance on various classification tasks, however, we are more interested to compare with recently proposed state-of-the-art models related to malware detection tasks specifically, as in (Athiwaratkun & Stokes (2017) and Kolosnjaji et al (2016)).
>
> Reference:
> 1. Ben Athiwaratkun and Jack W Stokes. Malware classification with lstm and gru language models and a character-level cnn. In International Conference on Acoustics, Speech and Signal Processing (ICASSP), pp. 2482–2486. IEEE, 2017.
> 2. Bojan Kolosnjaji, Apostolis Zarras, George Webster, and Claudia Eckert. Deep learning for classification of malware system call sequences. In Australasian Joint Conference on Artificial Intelligence, pp. 137–149. Springer International Publishing, 2016.

---

### Official Review · AnonReviewer3 · 2018-11-04
**DRL for malware control**

**Rating:** 4
**Confidence:** 3

**Review:**

The paper proposes an approach to use deep reinforcement learning to halt execution in detecting malware attacks. The approach seems interesting, there are some problems.

1. There is no good justification of using DRL to the problem. Action space is only continue and halt. Besides there should be no effect to the result by the previous action. So I don't think DRL is a good selection.
2. Experiments are weak. There is no detailed comparison to other existing works. Only one dataset is used.

---

> ### Author Response · Authors · 2018-11-26
> **Response to AnonReviewer3**
>
> Thank you very much for reviewing the paper and your helpful feedback.
> 1.	First, it is very natural to have a very small action space in reinforcement learning. For example, in solving a maze problem using RL, the action space only includes “left”, “right”, “up” and “down” four actions. In learning to play blackjack game using reinforcement learning (Sutton & Barto (1998)), the action space only includes “fold” and “hit” two actions.
> Second, as given by the state definition in the paper, a previous action will affect the current state. For instance, if a “Halt” action is chosen at the last state, the current state will be the same as the last state (i.e. loop back to the last state). Comparatively, if a “continue” action is selected at the previous state, the current state will contain a new event; at the same time, the previous events’ information is stored in the histogram.
> A very important reason why we choose to use DRL instead of standard deep learning  or other supervised learning models for this problem is because there is no ground truth for the stop/continue decisions for an anti-malware engine. The engine can only learn “when to stop” based on the reward/penalty between its interaction with the input file. The engine will stop if it is confident enough to decide:
> a. The file is a malware or
> b. The file is benign, and the engine can stop wasting computational resources.
> This conclusion is gradually learned through the interaction between the engine and the input file by accumulating the confidence rewards (either positive or negative). The procedure is most suitable to be modeled by reinforcement learning, which can learn the action using accumulated information without a concrete ground truth.
> 2.	We were not clear in the earlier draft, but we have compared the proposed system to many different baselines. We have made the point clearer in the text. As shown in Figure 6 and 7, we have compared our model’s result with 5 baselines using the recently proposed state-of-the-art models for malware classification. In the revised draft, we added another experiment using the convolutional neural network (CNN) based classifier introduced in Kolosnjaji et al (2016) for malware classification task and another one for a single hidden layer neural network. It is shown that our DRL-based model outperforms all 7 baselines in terms of the true positive rate at the same false positive rate.
> Considering the uniqueness of this type of malware detection task, there is no public dataset available. Also due to the sensitivity of the malware information, companies do not share these datasets in order to prevent attackers from using the datasets to generate adversarial attacks in the wild.
>
> Reference:
> 1.	A Sutton, Richard S., and Andrew G. Barto. Introduction to reinforcement learning. Vol. 135. Cambridge: MIT press, 1998
> 2.	Bojan Kolosnjaji, Apostolis Zarras, George Webster, and Claudia Eckert. Deep learning for classification of malware system call sequences. In Australasian Joint Conference on Artificial Intelligence, pp. 137–149. Springer International Publishing, 2016.

---

### Official Review · AnonReviewer1 · 2018-11-07
**Possibly useful malware detector but unclear paper and uncharacterized black box labels in dataset**

**Rating:** 5
**Confidence:** 2

**Review:**

This paper attempts to train a predictor of whether software is malware. Previous studies have emulated potential malware for a fixed number of executed instructions, which risks both false negatives (haven’t yet reached the dangerous payload) and false positives (malware signal may be lost amidst too many other operations). This paper proposes using deep reinforcement learning over a limited action space: continue executing a program or halt, combined with an “event classifier” which predicts whether individual parts of the program consist of malware. The inputs at each time step are one of 114 high level “events” which correspond to related API invocations (e.g. multiple functions for creating a file). One limitation seems to be that their dataset is limited only to events considered by a "production malware engine", so their evaluation is limited only to the benefit of early stopping (rather than continuing longer than the baseline malware engine). They evaluate a variety of recurrent neural networks for classifying malware and show that all significantly underperform the “production antimalware engine”. Integrating the event classifier within an adaptive execution control, trained by DQN, improves significantly over the RNN methods.

It might be my lack of familiarity with the domain but I found this paper very confusing. The labeling procedure (the "production malware engine”) was left entirely unspecified, making it hard to understand whether it’s an appropriate ground-truth and also whether the DRL model’s performance is usable for real-world malware detection.

Also, the baseline models used an already fairly complicated architecture (Figure 3) and it would have been useful to see the performance of simple heuristics and simpler models.

---

> ### Author Response · Authors · 2018-11-26
> **Response to AnonReviewer1**
>
> Thank you very much for reviewing the paper and your helpful feedback.
> 1. We added a paragraph to Appendix A to explain the labeling procedure of the files which describes that the labels are production labels that are used for training production antimalware classifiers. We also updated Appendix A to indicate that these labeled files contain the system calls generated by running the file in a production anti-malware emulator in the company’s production environment. We can move this section earlier in the paper if it is okay to go beyond 8 pages.
> 2. We added two more baselines for the file classifier in Figure 6. One is trained using a simple neural network model with only one hidden layer, and the other is trained using a conventional neural network (CNN) based model introduced in Kolosnjaji et al. (2016)  for malware classification. The original 5 baseline models are recently proposed state-of-the-art deep sequence-based models for malware detection.
>
> Reference:
> 1. Bojan Kolosnjaji, Apostolis Zarras, George Webster, and Claudia Eckert. Deep learning for classification of malware system call sequences. In Australasian Joint Conference on Artificial Intelligence, pp. 137–149. Springer International Publishing, 2016.

---

### Meta-Review · Area_Chair1 · 2018-12-13
**Interesting application of DRL, but too confuse presentation and too little experimental for the task and results**

**Confidence:** 4
**Recommendation:** Reject

**Metareview:**

The paper trains a classifier to decide if a program is a malware and when to halt its execution. The malware classifier is mostly composed of an RNN acting on featurized API calls (events). The presentation could be improved. The results are encouraging, but the experiments lack solid baselines, comparisons, and grounding of the task usefulness, as this is not done on an established benchmark.